# On-Policy Fine-grained Knowledge Feedback for Hallucination Mitigation

## Abstract

Hallucination occurs when large language models (LLMs) exhibit behavior that deviates from the boundaries of their knowledge during the response generation process. Previous learning-based methods focus on detecting knowledge boundaries and finetuning models with instance-level feedback, but they suffer from inaccurate signals due to off-policy data sampling and coarse-grained feedback. In this paper, we introduce _Reinforcement Learning for Hallucination_ (RLFH), a fine-grained feedback-based online reinforcement learning method for hallucination mitigation. Unlike previous learning-based methods, RLFH enables LLMs to explore the boundaries of their internal knowledge and provide on-policy, fine-grained feedback on these explorations. To construct fine-grained feedback for learning reliable generation behavior, RLFH decomposes the outcomes of large models into atomic facts, provides statement-level evaluation signals, and traces back the signals to the tokens of the original responses. Finally, RLFH adopts the online reinforcement algorithm with these token-level rewards to adjust model behavior for hallucination mitigation. For effective on-policy optimization, RLFH also introduces an LLM-based fact assessment framework to verify the truthfulness and helpfulness of atomic facts without human intervention. Experiments on HotpotQA, SQuADv2, and Biography benchmarks demonstrate that RLFH can balance their usage of internal knowledge during the generation process to eliminate the hallucination behavior of LLMs.

## 1 Introduction

Large Language Models have demonstrated remarkable capabilities in generating fluent and plausible responses. However, these models occasionally incorporate fabricated facts in truthful content, referred to as _hallucination_. For instance, as shown in Figure 1, the response of LLMs about "Turing" contains erroneous factual information, such as stating he was born in 1911 and was American.

The crux of hallucination is _the misalignment of the models' generation and their internal knowledge_ (Xu et al., 2024). This occurs when large language models produce behavior that does not align with the boundaries of their knowledge during the response generation process. Such misalignment leads to various hallucinatory behaviors in the response, including misleading responses, reckless attempts, and evasive ignorance. Specifically, **misleading responses** refers to instances where the model inaccurately answers a question within its knowledge boundary. **Reckless attempts**, on the other hand, occur when the model attempts to respond to a query beyond its knowledge scope. **Evasive ignorance** is when the model, despite possessing the necessary knowledge, refrains from providing an answer. Unfortunately, due to the opaque nature of model knowledge, we can only observe erroneous responses generated by large models or their refusal to respond, without accurately determining whether they have experienced hallucinations and the specific types of hallucinations they have experienced. Different types of hallucinations may even appear simultaneously within a single response. This immeasurable characteristic of hallucinations presents a significant challenge for mitigating them in LLMs.

To address the hallucination problem in large language models, previous work can be categorized into two directions: learning-based and editing-based. Learning-based methods involve detecting the model's knowledge boundaries and then fine-tuning it with feedback data to eliminate hallucinations. However, these methods suffer from off-policy data sampling (Zhang et al., 2024; Wan et al., 2024;

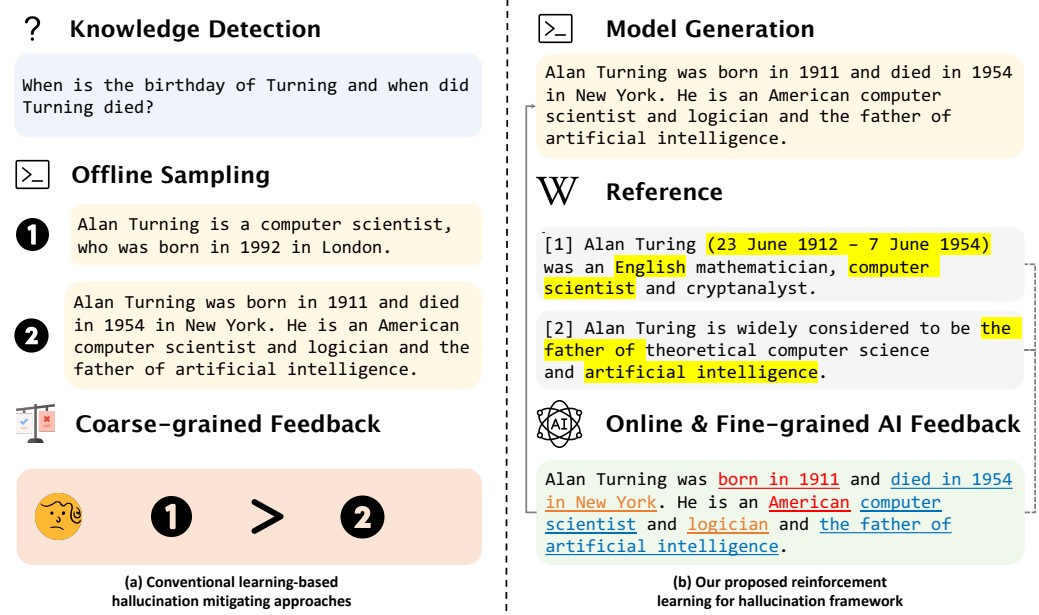

Figure 1: The figure illustrates the hallucinatory case and several hallucination mitigation methodologies. The factual information within the text is underlined. False content is highlighted in red, whereas accurate facts are indicated in blue. Statements with uncertain veracity are marked in orange.

Lin et al., 2024), leading to distribution shifts and suboptimal strategies. Besides, the instance-level, coarse-grained feedback (Sun et al., 2022; Tian et al., 2023; Kang et al., 2024) signals can not precisely pinpoint the content and type of hallucinations, potentially causing side effects. For example, a single sample might contain both correct and incorrect knowledge, and a coarse-grained feedback signal can optimize both of them in the same direction. Besides, due to the current lack of understanding of how models learn and express knowledge, existing knowledge detection techniques (Zhang et al., 2023a; Cheng et al., 2024; Yang et al., 2023) might be inaccurate. The detection outcomes can vary significantly based on specific prompts, and the data constructed based on model knowledge detection might not align with the real knowledge boundary of LLMs. In contrast, editing-based methods (Gou et al., 2023; Manakul et al., 2023) generate content first by LLM, and then edit it based on external ground-truth knowledge. However, considering the limitation of external knowledge, editing-based methods can only provide incremental improvements without fundamentally correcting the hallucination generation behavior of LLMs. Therefore, the key to solving the hallucination problem lies in providing fine-grained feedback signals tailored to the online policy LLM, enabling the model to explore its knowledge boundaries effectively and learn reliable generation behavior.

In this paper, we present *Reinforcement Learning for Hallucination* (RLFH), an online reinforcement learning approach predicated on fine-grained feedback for hallucination mitigation. The main idea behind RLFH is to enable LLMs to explore the boundaries of their internal knowledge and provide on-policy, fine-grained feedback on these explorations. This allows LLMs to effectively learn how to balance their usage of internal knowledge during the generation process, thereby eliminating the hallucination behavior of LLMs. To this end, RLFH first requires LLMs to explore potential outcomes and decompose the response into atomic facts. These atomic facts then undergo an assessment to verify their truthfulness and helpfulness by comparing the generated facts with ground truth from external knowledge sources. Then starting from these fine-grained, statement-level assessments, RLFH will trace back the judgment on atomic facts to the original outcomes, and construct token-level dense reward signals which can be directly leveraged to optimize the policy model. Finally, we adopt the online reinforcement algorithm with these token-level reward signals to adjust model behavior for hallucination mitigation. Furthermore, to address the need for timely and low-cost construction of reward signals in the on-policy optimization process, RLFH also introduces an LLM-based fact assessment framework to verify the truthfulness and helpfulness of atomic facts without human intervention. Specifically, we utilize LLMs to automatically determine whether an atomic fact aligns

with the facts described in the ground-truth document and assess the informativeness of the fact. This automated fact assessment ensures that reward signals can be obtained in real-time and accurately, thereby enabling RLFH to be directly used for on-policy behavior optimization.

We conducted experiments on HotpotQA, SQuADv2, and Biography benchmarks to evaluate the effectiveness of RLFH on hallucination mitigation. Results demonstrate that RLFH can effectively improve the truthfulness and informativeness of the model response. Compared with the initial model, the model after RLFH has achieved a significant improvement (+17.9% FactScore on average). Furthermore, compared to previous learning-based hallucination mitigation methods, RLFH can better mitigate hallucination behavior through on-policy and fine-grained signals, thus achieving an improvement (+2.0% FactScore on average).

To sum up, the primary contributions of this paper are threefold:

**1)** We propose Reinforcement Learning for Hallucination, an online reinforcement learning framework designed for mitigating hallucinations in large language models.

**2)** We propose to construct fine-grained, token-level knowledge feedback signals based on atomic fact judgment. By decomposing the outcomes of large models into atomic facts, providing statement-level evaluation signals, and tracing back the signals to the tokens of the original outcomes, we effectively achieve fine-grained feedback and learning for hallucination behaviors of LLMs.

**3)** We propose an LLM-driven method for evaluating the truthfulness and helpfulness of facts. By automatically comparing with ground truth documents, this method effectively constructs on-policy feedback signals automatically without human intervention.

## 2 RELATED WORKS

### 2.1 HALLUCINATION MITIGATION

Prior research (Zhang et al., 2023c; Ye et al., 2023; Tonmoy et al., 2024) has been dedicated to addressing the hallucination of Language Language Models. Some studies focus on reducing errors (Wang, 2019; Parikh et al., 2020) and supplementing missing knowledge (Ji et al., 2023) during data curation. Other works mitigate hallucination in either pre- or post-generation by retrieving external knowledge (Peng et al., 2023; Li et al., 2023b; Gou et al., 2023) or by exploiting self-consistency (Manakul et al., 2023; Shi et al., 2023; Lee et al., 2023). Recent studies put efforts into investigating the essence of the hallucination (Yu et al., 2024; Jiang et al., 2024) and resort to improving the model's factuality during the alignment stage. These works focus on resolving the inconsistency between the model's generation pattern and its internalized knowledge (Xu et al., 2024) through knowledge detection and coarse-grained feedback. Typically, these works attempt to delineate the boundary of model knowledge through explicit prompting (Zhang et al., 2023a; Yang et al., 2023; Cheng et al., 2024; Wan et al., 2024), self eliciting (Chen et al., 2024a; Lin et al., 2024), self-evaluation (Zhang et al., 2024) or by probing the model's internal states (Liang et al., 2024). Based on the detection of model knowledge, the data is meticulously crafted to align with the model's knowledge scope. Subsequently, the model is fine-tuned with coarse-grained feedback, which inspects the truthfulness of the response as a whole (Sun et al., 2022; Tian et al., 2023; Kang et al., 2024).

### 2.2 REINFORCEMENT LEARNING FROM HUMAN FEEDBACK

Reinforcement Learning from Human Feedback (Stiennon et al., 2020; Ouyang et al., 2022) has emerged as a noteworthy approach reaching significant success in the domain of the Language Language Model. Given the instability of reinforcement learning, some research (Lu et al., 2022; Rafailov et al., 2023; Dong et al., 2023) has attempted to learn human preferences directly from labeled data. In addition to sparse rewards, some works have resorted to designing more instructive rewards. One line of works (Wu et al., 2023; Lightman et al., 2023; Chen et al., 2024b; Cao et al., 2024) is dedicated to the acquisition of dense rewards. Another line of work (Ramé et al., 2023; Eisenstein et al., 2023; Coste et al., 2024; Ramé et al., 2024) concentrates on ensemble multiple reward models. Despite the majority of research prioritizing helpfulness and harmlessness, few studies (Wu et al., 2023; Tian et al., 2023; Liang et al., 2024) have explicitly considered truthfulness.

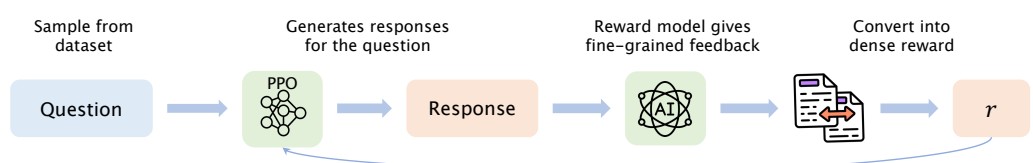

Figure 2: A diagram illustrating the steps of our algorithm: (1) Online sample response from tuning model, (2) Collect fine-grained knowledge feedback from AI-driven annotation pipeline and (3) Convert the language-form feedback into the token-level dense reward for reinforcement learning.

## 3 REINFORCEMENT LEARNING FOR HALLUCINATION

Given the train prompt set $\mathcal{X} = \{x_1, x_2, ..., x_{|\mathcal{X}|}\}$, the faithful reward model $\mathcal{M}$, and the reference document set $\mathcal{D} = \{d_1, d_2, ..., d_{\mathcal{D}}\}$, this section demonstrates how to mitigate the hallucination of the policy model $\pi$ via online fine-grained feedback reinforcement learning, as shown in Figure 2. Here's a detailed breakdown of each step: 1) **Response Generation**: Given the prompt $x_i$, the policy model $\pi$ generates a corresponding response $y_i$. This step involves the model using its current policy to produce an output based on the input prompt. 2) **Fine-grained Feedback from Faithful Reward Model** (§3.1): The faithful reward model $\mathcal{M}$ evaluates the generated response $y_i$ by comparing it with the reference document set $\mathcal{D}$. After the evaluation, the faithful reward model $\mathcal{M}$ provides fine-grained feedback $\mathcal{E}$ at the statement level. 3) **On-Policy Optimization with Token-level Reward** (§3.2): The detailed feedback $\mathcal{E}$ is translated into token-level rewards $r$. These rewards are then used to update the policy model $\pi$ using Proximal Policy Optimization (PPO), ensuring that the model learns to reduce hallucinations effectively.

### 3.1 FINE-GRAINED FEEDBACK FROM FAITHFUL REWARD MODEL

Given the prompt $x_i$ and its corresponding response $y_i$, the faithful reward model $\mathcal{M}$ will give fine-grained feedback concerning the truthfulness and helpfulness at the statements-level granularity. Specifically, the faithful reward model $\mathcal{M}$ initially extracts atomic statements $\mathcal{E}_i = \{e_1, e_2, ..., e_{|\mathcal{E}_i|}\}$ from the response $y_i$, where each statement $e_j$ represents an atomic fact in the response. After that, the faithful reward model $\mathcal{M}$ further verifies each statement $e_i$ with the reference document and gives fine-grained feedback.

#### 3.1.1 STATEMENT EXTRACTION

Given a query $x$ and its corresponding output $y$, the reward model extracts valid factual statements in a hierarchical manner. Specifically, $\mathcal{M}$ initially divide the response into sentences $\{s_i\}_{i=1}^{M}$ and then extract all valid factual statements $\{e_{ij}\}_{j=1}^{N_i}$ from the each sentence $s_i$. There are two main reasons for the aforementioned hierarchical method: (1) Splitting the response into sentences before extracting statements consistently yields finer granularity; (2) Extracting statements sentence-by-sentence facilitates the conversion from language-form annotation to dense reward. To mitigate noise during the extraction process, we further filter out sentences without valid statements.

#### 3.1.2 FACTUAL VERIFICATION

The reward model $\mathcal{M}$ evaluates the truthfulness of the extracted factual statements by comparing them with external knowledge sources. We retrieve the relevant supporting materials $c_i^{L}_{i=1} \subset \mathcal{C}$ from the external reference document set $\mathcal{D}$ for each statement $e$. With these supporting contexts, the reward model $\mathcal{M}$ performs the statement verification as reading comprehension. Specifically, this process is represented as $k_{\text{truth}} = \mathcal{M}\left(e, \{c_i\}_{i=1}^{L}\right)$. However, the limited scope of supporting materials and the inherent unpredictability of model generation may lead to some statements lacking verifiable truthfulness. To mitigate this challenge, we introduce the "*Vague*" category for statement classification. Consequently, the model classifies each statement into the following labels: 1) *Correct*: statement supported by evidence; 2) *Hedged Correct*: accurate statement with uncertainty; 3) *Vague*:

Figure 3: A schematic representation of fine-grained feedback and token-level reward strategy methodology is presented. Initially, the statements are extracted in a hierarchical fashion. Subsequently, the veracity and utility of each statement are assessed. Ultimately, the structured feedback is mapped back into a dense reward via the Longest Common Subsequence (LCS) algorithm.

truthfulness uncertain; 4) *Hedged Wrong*: false statement with uncertainty; 5) *Wrong*: statement contradicted by evidence.

### 3.1.3 INFORMATIVENESS ASSESSMENT

In addition to assessing correctness, RLFH also evaluates the informativeness of the response. Each statement's helpfulness is rated on a five-point scale, from providing crucial information (+5) to containing useless details (+1). Unlike the individual verification of each statement in the statement verification process, assessing informativeness requires simultaneously evaluating multiple statements from the response. This is because when evaluating informativeness, we typically compare the information content between different statements to determine their relative importance or relevance. This requires considering the overall context and the comprehensiveness of the content, rather than just the truthfulness of individual statements. In this way, we assess multiple statements in a single-pass prompting, which denoted as $\{k_{\text{info}}^i\}_{i=1}^N = \mathcal{M}\left(\{e_i\}_{i=1}^N\right)$. The introduction of informativeness prevents the trivial hack that the model either rejects the majority of responses or produces only brief answers, both of which are undesirable outcomes.

### 3.2 ON-POLICY OPTIMIZATION WITH TOKEN-LEVEL REWARD

Given the fine-grained, statement-level feedback from the reward model, RLFH will trace back the judgment on atomic facts to the original response, and construct token-level dense reward signals, which can be directly leveraged to optimize the policy model. Finally, we adopt the online reinforcement algorithm with these token-level reward signals to adjust model behavior for hallucination mitigation.

### 3.2.1 DENSE REWARD CONVERSION

We represent the helpfulness and truthfulness of the response through the dense reward conversion presented in Figure 3. Due to the mutually exclusive nature (Xu et al., 2024) of these two objectives, the model learns to balance the pursuit toward two dimensions, thereby acquiring the appropriate strategy for the utilization of their internalized knowledge.

**Truthfulness** For each extracted statement, we will assign a truthfulness reward computed as follows:

$$r_{\text{truth}} = \alpha f(k_{\text{truth}})|g(k_{\text{info}})| \tag{1}$$

where $f$ and $g$ represent manually designed functions that transform the labels $k$ into scalar values. In principle, $f$ gives a positive reward to the truthful statement and a negative reward to the unverifiable or false statement. Due to the phenomenon of hallucination snowball (Zhang et al., 2023b), i.e. some critical errors lead to the magnification of the hallucinations, $g$ is included to diversify the importance of different statements. The sign of the output function $f$ is maintained by passing the outcome of the function $g$ through an absolute value function. The coefficient $\alpha$ balances between the truthfulness reward and the helpfulness reward.

The reward $r_{\text{truth}}$ will then be mapped back to the token sequences of the model's responses $y$ through a hierarchical structure constructed in prior annotations. Specifically, we first employ the Longest

Common Subsequence algorithm to map the characters of each statement $e_{ij}$ back to its originating sentence $s_i$. Subsequently, each sentence $s_i$ is mapped back to the model's response $y$ through the Longest Common Substring algorithm. Finally, the reward $r_{\text{truth}}$ is assigned to the token in the response which corresponds to the index of the last character in the statement.

**Informativeness** For each sentence, we assign an informative reward based on the statements encompassed as follows:

$$r_{\text{info}} = \beta \log \left( 1 + \max \left( \epsilon, \sum_i^N g\left(k_{\text{info}}^i\right) \right) \right) \tag{2}$$

In this equation, $N$ denotes the total number of statements within a sentence, while $\epsilon$ represents the minimum reward threshold serving to penalize non-informative statements. As indicated by the equation, the reward increases with the number of statements in a sentence and their respective informativeness. However, the rate of growth of the reward diminishes rapidly. Conversely, the penalty for producing non-informative statements by the model escalates swiftly. We utilize the same method as the correctness reward to map the reward value back to the response token sequence.

### 3.2.2 ONLINE REINFORCEMENT LEARNING

Given our reward function, the training process is to maximize the following objective:

$$\arg\max_\pi \mathbb{E}_{x\sim\mathcal{X}, y\sim\pi} \left[ \sum_{i=1}^T r\left(y_t, (x, y_{1:t})\right) \right] \tag{3}$$

The policy model $\pi$ is optimized through online reinforcement learning (Tang et al., 2024). Specifically, we first sample the prompt $x$ and responding response $y$. Then our fact assessment framework provides fine-grained feedback and converts it into token-level dense reward $r = [r_1, r_2, ..., r_T]$, where $T$ stands for the total length of the response $y$. Given the timely nature of the LLM-based fact assessment framework, the reward can be collected online. Finally, the model $\pi$ is optimized by the Proximal Policy Optimization (PPO) (Schulman et al., 2017) algorithm.

## 4 EXPERIMENT

### 4.1 SETTINGS

**Datasets** We employ three distinct datasets for our experiments. Following the approach in Min et al. (2023), we filter out prompts lacking corresponding wiki pages for both training and evaluation. Additionally, we sample 20,000 questions from **HotpotQA** (Yang et al., 2018) and utilize the English Wikipedia from 04/01/2023 as the retrieval corpus for training. We filtered the questions in the Hotpot QA with less than 5 words and sampled 256 questions for evaluation. We deduplicate the question in **SQuADv2** (Rajpurkar et al., 2016) with the same reference wiki pages, leaving 191 questions for out-of-distribution QA evaluation. **Biography** is the identical biographies dataset as utilized in the FactScore (Min et al., 2023) for evaluation out-of-distribution of different forms.

**Baselines** We compare RLFH with two different types of baselines: 1) *hallucination mitigation methods* using the same initialize model, including inference-time intervention (**INI**) (Li et al., 2023a), decoding by contrasting layers (**DOLA**) (Chuang et al., 2023) and finetuning for factuality (**FACT**) (Tian et al., 2023); 2) *advanced aligned models* with the same model size of our model (7B), including **Zephyr** (Tunstall et al., 2023), **Orca** (Mukherjee et al., 2023), **Llama2 Chat** (Touvron et al., 2023), and **Vicuna 1.5** (Zheng et al., 2023).

**Evaluation** To evaluate the truthfulness and helpfulness of each generated response, we employ the FactScore (Min et al., 2023) pipeline run by GPT4 (OpenAI, 2023). FactScore pipeline extracts the facts and determines the correctness of each fact. For each dataset, we report the number of correct and relevant facts (# Cor.), the number of inaccurate facts (# Inc.), the ratio of responded questions (% Res.), and the computed FactScore metrics (Score.).

**Implementation** Our training implementations are developed based on TRLX (Havrilla et al., 2023). The base model utilized is Vicuna-7b-1.5 (Zheng et al., 2023) and Mixtral-8x7B-Instruct (AI, 2023) is deployed to provide fine-grained AI feedback. Detailed prompts are shown in Appendix 6.1.

Table 1: Experiment results on HotpotQA, SQuADv2, and Biography.

| Model | HotpotQA | | | | SQuADv2 | | | | Biography | | | |
|---|---|---|---|---|---|---|---|---|---|---|---|---|
| | #Cor. | #Inc. | %Res. | Score | #Cor. | #Inc. | %Res. | Score | #Cor. | #Inc. | %Res. | Score |
| *Advanced Aligned Model* | | | | | | | | | | | | |
| Orca2 | 20.04 | 9.042 | 0.996 | 0.467 | 22.04 | 11.57 | 0.995 | 0.580 | 11.99 | 33.86 | 1.000 | 0.264 |
| Zephyr | 10.48 | 6.191 | 0.965 | 0.610 | 14.34 | 8.727 | 0.995 | 0.657 | 19.85 | 41.20 | 1.000 | 0.325 |
| Llama2 | 12.71 | 12.23 | 0.922 | 0.544 | 24.90 | 21.31 | 0.990 | 0.556 | 23.08 | 58.50 | 0.978 | 0.288 |
| Vicuna | 7.430 | 6.320 | 0.910 | 0.569 | 13.07 | 6.597 | 0.979 | 0.669 | 15.73 | 27.71 | 0.830 | 0.347 |
| *Hallucination Mitigation methods based on Vicuna-7B* | | | | | | | | | | | | |
| ITI | 21.66 | 17.92 | 0.961 | 0.547 | 32.49 | 26.37 | 0.990 | 0.544 | 21.58 | 47.93 | 0.989 | 0.311 |
| DOLA | 6.734 | 5.688 | 0.801 | 0.582 | 12.78 | 7.644 | 0.963 | 0.647 | 13.25 | 24.16 | 0.681 | 0.357 |
| FACT | 13.31 | 7.363 | 0.945 | 0.647 | 16.13 | 7.931 | 0.984 | 0.676 | 16.46 | 20.75 | 0.736 | 0.457 |
| RLFH | 13.05 | 8.304 | 0.645 | **0.655** | 23.40 | 10.96 | 0.953 | **0.683** | 18.08 | 19.20 | 0.692 | **0.474** |

## 4.2 MAIN RESULTS

Table 1 presents the performance of all baselines and RLFH on three datasets. We can see that:

**1. Our method significantly mitigates hallucination.** As demonstrated in Table 1, our method achieved the highest FactScore across all datasets. Given that FactScore is a well-established metric for assessing the factuality of long-form generation with the support of external knowledge, we argue that the improvement substantiates the effectiveness of our algorithm in mitigating hallucination.

**2. The improvement is generalizable to out-of-distribution prompts.** Notably, despite being trained on the HotpotQA dataset, our algorithm demonstrated improved accuracy on two out-of-distribution datasets of different task settings. This indicates the reasonable utilization of knowledge learned by our algorithm is a meta-ability that can be generalized.

**3. The aligned model is more conservative but provides more information within its capacity.** As shown in Table 1, our trained model decreases in response ratio. This can be ascribed to two primary factors: (1) The FactScore pipeline determines whether the model refuses to answer by detecting the presence of characteristic words, while our trained model tends to express uncertainty even when providing relevant information, leading to an underestimation of the indicators. (2) Our tuned model is overall more conservative, as evidenced by the decreased response ratio across all datasets. Even though, the model's responses generally contain more statements, indicating increased confidence in answering questions deemed answerable. Also, the model refuses more questions on the HotpotQA and Biography datasets and less on the SQuADv2 dataset. This can be attributed to that SQuADv2 is comparatively easier than the other two datasets as indicated in the metrics of the base model.

## 4.3 DETAILED RESULTS

To investigate the changes in model behavior after RLFH, we conducted a detailed analysis of 5000 questions from HotpotQA that were not included in the training phase.

**1. Our method augments the ratio of high-accuracy responses.** As indicated in Figure 4, the distribution of average statement accuracy shifts significantly towards higher accuracy, signifying a reduction in responses with lower accuracy and an increase in responses with higher accuracy. It is noteworthy that there is a significant increase in responses with an accuracy exceeding 0.7, suggesting the model provides more reliable responses after training.

**2. Our algorithm enhances the responses' helpfulness while suppressing errors.** As illustrated in Figure 6a, our algorithm increases the proportion of responses containing more statements. As specified in Figures 6b and 6c, this is achieved by increasing the responses containing more correct statements, while concurrently reducing the responses containing incorrect statements. Figure 5 jointly estimates the truthfulness and helpfulness of the model's responses. As presented in the figure, the distribution shifts toward the upper right direction, indicating that the model tends to generate more informative responses while minimizing the occurrence of unverifiable statements.

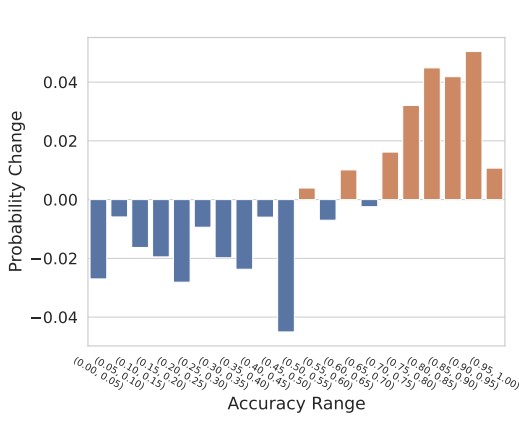

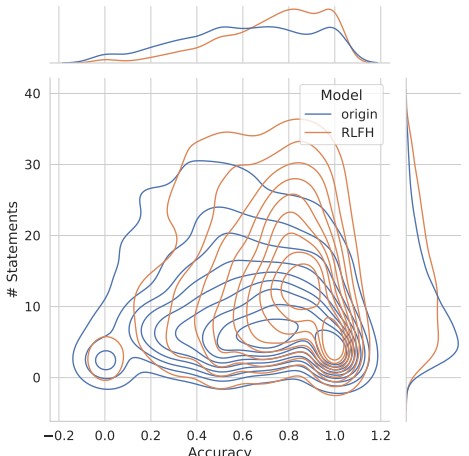

Figure 4: The statement accuracy distribution difference before and after training. The distributions are normalized due to the filtering of rejected responses resulting in different numbers of prompts.

Figure 5: The statement's accuracy and the number of statements per response distribution of the responses pre- and post-training.

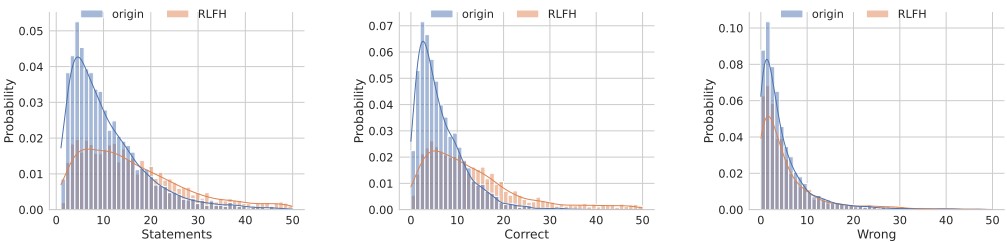

(a) Distribution of the number of statements per response

(b) Distribution of the correct statements per response

(c) Distribution of the wrong statements per response

Figure 6: The distributions of the quantity of the statements per response.

**3. Our algorithm aligns the model's behavior with its knowledge boundary.** We compare the distribution of the number of statements under varying accuracy ranges in both pre- and post-training. As illustrated in Figure 7, there is a noticeable shift in the model's behavioral strategy as the accuracy range changes from low to high. Specifically, the number of statements tends to decrease in the lower accuracy range and increase in the higher accuracy range. This suggests that the model is learning to provide information in a manner that is commensurate with its knowledge. Furthermore, we inspect the relationship between the refusal ratio and the original response accuracy. We determine the refusal by our annotation pipeline for the reason mentioned in Section 4.2. As shown in Figure 8, the model tends to refuse questions that it originally performed poorly. This is expected as the model is penalized more frequently for attempting to answer uncertain questions.

## 4.4 IMPACT OF REWARD GRANULARITY

In this section, we conduct an ablation experiment to investigate the impact of the granularity of the reward on the performance. Specifically, we evaluate the three different granularities of reward signals: response-level, sentence-level, and statement-level. The statement-level reward is the default setting described in previous sections. For the sentence-level reward, feedback for each sentence is incorporated into a single reward assigned at the end token of each sentence. For the response-level reward, feedback is aggregated into a single value representing the overall quality. As illustrated in Table 2, the statement-level reward achieved the highest FactScore, indicating that more fine-grained feedback tends to yield better performance. Note that although the model under statement-level

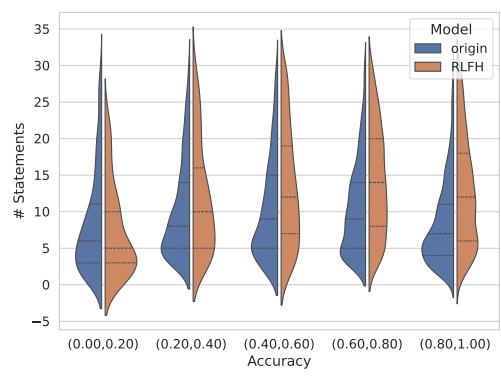 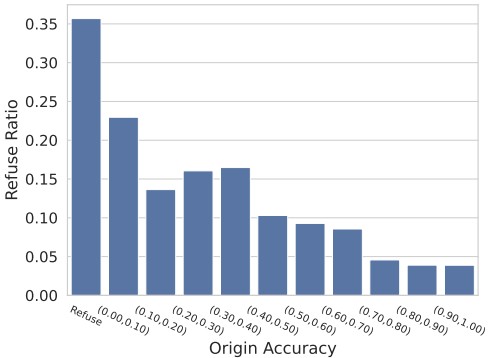

Figure 7: The distribution of the number of statements under different statement accuracy ranges.

Figure 8: The ratio of prompts rejected after training, grouped by their original accuracy levels.

reward had the lowest response ratio, it reached the highest number of statements per response. This phenomenon could be a result of the underestimation problem discussed before.

### 4.5 IMPACT OF ANNOTATION MODEL

In this section, we explore the impact of the annotation model used for collecting feedback. We employ open-source models of varying sizes to execute the annotation pipeline. As depicted in Table 3, many open-source models are sufficiently capable of delivering beneficial factual supervision signals. Note that different models exhibit distinct characteristics when completing these tasks, e.g. the granularity of statement extraction, the inclination towards informative assessment, or the ability to evaluate factuality within a supporting context. These differences lead to varying model behaviors. For instance, even though our model supervised by the same SFT model achieves the highest FactScore, it is at the cost of helpfulness in the measurement of the average statements contained in the response.

Table 2: Results of our algorithm under reward granularity levels on HotpotQA.

| Reward | HotpotQA | | | |
|---|---|---|---|---|
| | #Cor. | #Inc. | %Res. | Score. |
| Paragraph | 7.152 | 5.660 | 0.867 | 0.639 |
| Sentence | 11.17 | 6.453 | 0.715 | 0.645 |
| Statement | 13.05 | 8.304 | 0.645 | **0.655** |

Table 3: Results of our algorithm under reward models powered by different LLMs on HotpotQA.

| Model | HotpotQA | | | |
|---|---|---|---|---|
| | #Cor. | #Inc. | %Res. | Score. |
| Vicuna-7b | 4.207 | **2.023** | 0.800 | **0.697** |
| Vicuna-13b | 9.371 | 4.637 | 0.773 | 0.668 |
| Llama2-70b | 9.292 | 4.675 | 0.781 | 0.672 |
| Falcon-30b | 5.148 | 3.156 | 0.777 | 0.652 |
| Mixtral | **13.05** | 8.304 | 0.645 | 0.655 |
| GPT4[1] | 5.590 | 3.648 | 0.750 | 0.638 |

### 5 CONCLUSION

In this work, we introduce *Reinforcement Learning for Hallucination* (RLFH), a fine-grained feedback-based online reinforcement learning method for hallucination mitigation. RLFH enables LLMs to explore their knowledge scope and adjust their behavior based on fine-grained on-policy feedback. Specifically, our approach provides fine-grained knowledge feedback based on atomic fact judgment and constructs token-level dense rewards for online reinforcement learning. Experiment results on three factual benchmarks show that RLFH can significantly improve the truthfulness and informativeness of LLMs under both in-distribution and out-of-distribution settings. For future work, we plan to extend our method to mitigate hallucinations in multi-modal large language models.

---

[1]Due to the cost of API calls, single-pass prompting discussed in the Appendix 6.1 is used for the annotation.

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

## 6 APPENDIX

### 6.1 PROMPT FOR AI FEEDBACK

The rollout stage in reinforcement learning requires online sampling, thus the annotation procedure must concurrently consider efficiency and effectiveness. Given the time-intensive nature of this process, we have implemented a pipeline prompting strategy. The entire procedure is divided into extraction, verification, and assessment, responding to the descriptions in Section 3.1, using prompts in Table 5, 6, and 7, respectively. As discussed in Section 4.5, most open-source models with certain capabilities in instruction following and reading comprehension are able to accomplish these tasks.

Due to the high cost and low throughput of the GPT4 API call, we opted for another single-pass prompting pipeline to complete the annotation, as illustrated in Table 4. The primary distinction between the single-pass and pipeline strategies lies in the granularity of the extracted statements. The pipeline strategy tends to fragment sentences in a more detailed manner resulting in more statements. We find the majority of open-source models failed to complete single-pass prompting satisfactorily.

### 6.2 CASE STUDY

In order to get more intuitive insights into our algorithm, we illustrate the evolution of the model's response throughout the training process. Two representative cases are presented in Table 8. The first case demonstrates the algorithm's capability to diminish the number of incorrect statements within the response. This is accomplished by adopting a more conservative approach towards unfamiliar questions or by simply declining to provide an answer. In the second case, our algorithm amends the model's response from being entirely incorrect to almost correct. This result is encouraging as the test question is not included in the training set, indicating that the model is learning to deliver more accurate and effective responses by better leveraging its inner knowledge, rather than by acquiring additional knowledge from the training corpus. The cases imply that in addition to inadequate knowledge, the model's inability to reasonably utilize knowledge is also a nonnegligible factor contributing to hallucinations.

Table 4: The prompt used for single-pass annotation.

Check the factuality and helpfulness of a response to a question based on the materials.
- Extraction: Break the sentences with object facts into atomic statements.
- Truthfulness Verification: verify each statement based on the materials.
    - "Correct": The statement is proved by the materials.
    - "Hedged correct": The statement is expressed with uncertainty but is true.
    - "Vague": The statement can not be decided as true or false based on the materials.
    - "Hedged wrong": The statement is expressed with uncertainty and is false.
    - "Wrong": The statement is denied by the materials.
- Informativeness Assessment: evaluate the helpfulness of each statement.
    - "5": The statement is an answer to the question.
    - "4": The statement provides crucial information.
    - "3": The statement gives relevant content.
    - "2": The statement contains supplementary facts.
    - "1": The statement is repeating information, providing useless information.
- Details:
    - Do not consider the correctness of the statement when assessing informativeness.
    - Follow the valid JSON format **(Use escape characters, e.g. " for ")** shown in the example.
    - Skip the sentences like "It's hard to say ...", "I'm sorry, I don't know ...". If there is no valid statement, output {}

>> Example:
[Materials]
- First for Women is a women's magazine published by Bauer Media Group in the USA. The magazine was started in 1989. It is based in Englewood Cliffs, New Jersey. In 2011 the circulation of the magazine was 1,310,696 copies."
- Arthur's Magazine (1844–1846) was an American literary periodical published in Philadelphia in the 19th century. Edited by T.S. Arthur, it featured work by Edgar A. Poe, J.H. Ingraham, Sarah Josepha Hale, Thomas G. Spear, and others. In May 1846 it was merged into "Godey's Lady's Book".
- Which magazine was started first Arthur's Magazine or First for Women? The correct answer may be "Arthur's Magazine".
[Question]
Which magazine was started first, Arthur's Magazine founded by Arthur K. Watson, or First for Women?
[Response]
It is difficult to say which game has been released in more versions without more information, so I can only guess based on my training data. Arthur's Magazine was likely started first. It was possibly founded in 1923 by Arthur K. Watson, a prominent publisher in the field of men's magazines. First for Women, on the other hand, was not founded until 1989. It was created as a spin-off of Family Circle magazine, which was founded in 1957.
[Annotation]
{"Arthur's Magazine was likely started first.": {"Arthur's Magazine was likely started first.": ["Correct", 5]}, "It was possibly founded in 1923 by Arthur K. Watson, a prominent publisher in the field of men's magazines.": {"It was possibly founded in 1923.": ["Wrong", 4], "It was founded by Arthur K. Watson.": ["Wrong", 3], "Arthur K. Watson is a prominent publisher in the field of men's magazines.": ["Vague", 2]}, "First for Women, on the other hand, was not founded until 1989.": {"First for Women was not founded until 1989.": ["Correct", 4]}, "It was created as a spin-off of Family Circle magazine, which was founded in 1957.": {"It was created as a spin-off of Family Circle magazine.": ["Vague", 3], "Family Circle magazine was founded in 1957.": ["Vague", 2]}}

>> Real Problem:

Table 5: The prompt used for extracting statements.

---

- Find every sentence containing object facts.
- Break sentences into atomic statements.
- If there is no valid sentence, output "No statements".
- Skip the sentences without statements.
- Do not output any explanation or other words.
- Strictly follow the output format shown in the example.

Here is an example:
# Response
It is difficult to say which game has been released in more versions without more information, so I can only guess based on my training data. Arthur's Magazine was likely started first. It was possibly founded in 1923 by Arthur K. Watson, a prominent publisher in the field of men's magazines. First for Women, on the other hand, was not founded until 1989. It was created as a spin-off of Family Circle magazine, which was founded in 1957.
# Statements
>> Sentence 1: Arthur's Magazine was likely started first.
* Arthur's Magazine was likely started first.
>> Sentence 2: It was possibly founded in 1923 by Arthur K. Watson, a prominent publisher in the field of men's magazines.
* Arthur's Magazine was possibly founded in 1923.* Arthur's Magazine was founded by Arthur K. Watson.
* Arthur K. Watson is a prominent publisher in the field of men's magazines.
>> Sentence 3: First for Women, on the other hand, was not founded until 1989.
* First for Women was not founded until 1989.
>> Sentence 4: It was created as a spin-off of Family Circle magazine, which was founded in 1957.
* First for Women was created as a spin-off of Family Circle magazine.
* Family Circle magazine was founded in 1957.

And then comes your task:

---

Table 6: The prompt used for verifying statements.

---

Choose from "Correct", "Vague" and "Wrong" for the verification of the statement.
- "Correct": The statement is supported by the materials.
- "Vague": Hard to determine the truthfulness of the statement based on the materials.
- "Wrong": The statement is negated by the materials.
Directly output the verification result without explanation.

Here is an example:
# Materials
- First for Women is a women's magazine published by Bauer Media Group in the USA. The magazine was started in 1989. It is based in Englewood Cliffs, New Jersey. In 2011 the circulation of the magazine was 1,310,696 copies."
- Arthur's Magazine (1844–1846) was an American literary periodical published in Philadelphia in the 19th century. Edited by T.S. Arthur, it featured work by Edgar A. Poe, J.H. Ingraham, Sarah Josepha Hale, Thomas G. Spear, and others. In May 1846 it was merged into "Godey's Lady's Book".
- The correct answer for the question "Which magazine was started first Arthur's Magazine or First for Women" may be "Arthur's Magazine".
# Statement
Arthur's Magazine was likely started first.
# Verification
Correct

And then comes your question:

---

Table 7: The prompt used for assessing the importance of the statement.

---

Evaluate the helpfulness of the statement:
- "5": The statement answer the question.
- "4": The statement provides crucial information.
- "3": The statement contains relevant facts.
- "2": The statement is about other supplementary facts.
- "1": The statement is useless or not relevant at all.
Directly output the evaluation result without explanation.

Here is an example:
# Question
Which magazine was started first Arthur's Magazine founded by Arthur K. Watson or First for Women?
# Response
It is difficult to say which game has been released in more versions without more information, so I can only guess based on my training data. Arthur's Magazine was likely started first. It was possibly founded in 1923 by Arthur K. Watson, a prominent publisher in the field of men's magazines. First for Women, on the other hand, was not founded until 1989. It was created as a spin-off of Family Circle magazine, which was founded in 1957.
# Statement
Arthur's Magazine was possibly founded in 1923.
# Evaluation
4

And then comes your task:

---

Table 8: The table provided delineates the cases encountered during the training process. Correct segments are highlighted in green, while those containing vague information are denoted in brown. Incorrect segments are marked in red. The reward is the aggregation of the rewards of each token.

| Prompt | Response | Reward |
|---|---|---|
| "Tina in the Sky with Diamonds" is the second episode of the fifth season of which television? | "Tina in the Sky with Diamonds" is the second episode of the fifth season of the television series "Breaking Bad". It was written by series creator Vince Gilligan and directed by Michelle MacLaren. The episode aired on August 10, 2014. | -5.696 |
| | "Tina in the Sky with Diamonds" is the second episode of the fifth season of the television series "The Crown," which was created by Peter Morgan. | -2.066 |
| | I'm sorry, but I am not able to provide information about specific episodes of television series as my training data only goes up until 2021 and I don't have access to any information about future episodes. | 0 |
| Are Veratrum and Leontopodium genuses of plants that are found in the same family? | Yes, Veratrum and Leontopodium are both genera of plants that are found in the same family, the Veratrum family (Veratrumaceae). | 0.4555 |
| | Veratrum and Leontopodium are not genuses of plants that are found in the same family. Veratrum is a genus of plants belonging to the family Meliaceae, while Leontopodium is a genus of plants belonging to the family Asteraceae. The two genera are not closely related and are found in different parts of the world. | 3.432 |
| | No, Veratrum and Leontopodium are not genera of plants that are found in the same family. They are both genera of plants, but they belong to different families. Veratrum is a genus of plants that belongs to the family Melanthiaceae. It contains several species of flowering plants that are native to North and South America, as well as parts of Asia. Leontopodium, on the other hand, is a genus of plants that belongs to the family Asteraceae. It contains several species of flowering plants that are native to the Alps and other mountain ranges in Europe. So, while both genera contain species of flowering plants, they are not closely related and belong to different families. | 16.653 |