# OpenReview forum: "On-Policy Fine-grained Knowledge Feedback for Hallucination Mitigation"
_ICLR.cc/2025/Conference — Submitted to ICLR 2025_

### Official Review · Reviewer_hGoa · 2024-11-05

**Soundness:** 3
**Presentation:** 3
**Contribution:** 2
**Rating:** 5
**Confidence:** 3

**Summary:**

This paper presents RLFH, a method that decomposes the output of large language models into atomic facts and validates them with reference documents. Then the sentence-level evaluation feedback is provided. After that, RLFH traces the feedback back to the original response and builds a token-level reward signal. Finally, an online reinforcement algorithm with these signals adjusts the model behavior to alleviate the illusion. The main contribution is to propose an online reinforcement algorithm based on fine-grained feedback signals and a non-manual, LLM-driven reward model.

**Strengths:**

1.The motivation of the paper is reasonable, and the problems to be solved are relatively important.

2.The model adjusts the model itself based on the evaluation feedback of the results, rather than just modifying the results. The method is interesting.

3.The analysis of the experimental results is detailed. Both the overall summary analysis and some detailed research are given.

**Weaknesses:**

1.The paper fails to fully solve the problem it raises. Evasive ignorance, for example, is a type of illusion that can't be detected or changed.

2.The model simply gives feedback on the correctness of each sentence based on the original answer. If the original answer itself has information missing, the model will not be able to correct and give feedback.

3. Please discuss the potential challenges or benefits of extending their method to other LLMs.

**Questions:**

For page 5 line 267: “Table ??” , please specify which table you are referring to.

---

> ### Author Response · Authors · 2024-11-25
>
> Thank you for your constructive feedback. For the mentioned weakness, we want to provide further clarification:
>
> ***Weakness 1:***
> Although our proposed method cannot completely resolve the hallucination problem—since theoretically, as noted in [1], it cannot be entirely eliminated—its primary aim, much like most current work [2, 3], is to mitigate hallucinations. Compared to other works, the RLFH algorithm leverages the fine-grained reward information provided by the LLM itself to achieve on-policy hallucination mitigation. Unlike some other methods that utilize off-policy approaches, the on-policy characteristic of RLFH better aligns the model's factual performance with its internal knowledge boundaries. For instance, regarding the evasive ignorance you mentioned, ideally, an on-policy algorithm would encourage the model to provide content it is most confident about, as this increases the expected overall reward.
> - [1] Hallucination is Inevitable: An Innate Limitation of Large Language Models
> - [2] Fine-tuning Language Models for Factuality
> - [3] FLAME: Factuality-Aware Alignment for Large Language Models
>
> ***Weakness 2:***
> When verifying the factuality of responses, RLFH does not rely solely on the original answer to the question. Instead, it can also utilize external knowledge sources for further validation (in our case, we use the Wiki knowledge base). When the external knowledge base does not cover the information contained in the response, we require the model to label the information as "Vague," indicating that its accuracy cannot be determined. Such information receives a lower reward than "Correct" information but a higher reward than clearly "Wrong" information. This approach encourages the model to avoid generating ambiguous information that is difficult to assess for accuracy. We believe that such content is also a form of hallucination.
>
> ***Weakness 3:***
> Our algorithm does not impose any competency requirements on the policy itself. However, for the reward model, we need it to possess some level of instruction-following capability to conduct factual verification according to our predefined prompts. As discussed in Section 4.5, we found that more capable models perform this task better. Furthermore, when considering the relationship between the capabilities of the policy and the reward model, we believe that our approach does not require the reward model to be more capable than the policy. This is because the reward model can rely on external knowledge for factual verification, which is a relatively straightforward task.
>
> ***Question 1:***
> Apologies for the confusion. We have rectified this issue in the new version of the paper.

---

> ### Author Response · Authors · 2024-11-28
>
> Dear Reviewer,
>
> Considering the rebuttal deadline is approaching, we sincerely hope to receive your response. If you have any further questions or concerns regarding our explanation, please do not hesitate to contact us. Your response is of great significance to us for improving this work, and we look forward to hearing from you.

---

### Official Review · Reviewer_rYQp · 2024-11-06

**Soundness:** 3
**Presentation:** 2
**Contribution:** 3
**Rating:** 5
**Confidence:** 3

**Summary:**

The paper introduces Reinforcement Learning for Hallucination (RLFH), an innovative approach aimed at reducing hallucination in large language models (LLMs). It focuses on instance-level feedback and relies on off-policy data sampling, RLFH provides fine-grained, on-policy feedback at the token level. This method decomposes responses into atomic facts and applies statement-level evaluations to assign feedback to specific tokens in the output, guiding the model to reduce hallucinations. To facilitate on-policy optimization, RLFH includes an automated fact assessment framework to evaluate the truthfulness and helpfulness of these facts. Experimental results on HotpotQA, SQuADv2, and Biography datasets show that RLFH effectively mitigates hallucination in LLMs by balancing their use of internal knowledge during response generation.

**Strengths:**

1.RLFH provides a novel approach to feedback by breaking down outputs into atomic facts and assigning token-level rewards, which allows for a more precise correction of hallucination errors.

2.By using on-policy data sampling, RLFH ensures that the feedback provided is more aligned with the model’s real-time behavior, improving the relevance and effectiveness of the adjustments made to the model.

3.the experimental results have shown the effectiveness of the proposed approach.

**Weaknesses:**

1.the proposed method is not a fully reliable on-policy method, as the feedback is provide by the LLM itself, not trustworthy feedbacks. Thus, it is hard to say after multi-turn iteration, if the model will be better, and maybe the wrong knowledge will accumulate.

2.the compared baselines should involve RLHF-based methods, i.e, DPO and RLAIF. Although these methods are not specially designed for on-policy optimization, it is necessary to show how they will perform under this occasion, to verify the motivation in the introduction part.

3.the writing of this paper needs to be polished. There are several typos: e.g., "Turning" -> "Turing"

**Questions:**

Please refer to the weaknesses

---

> ### Author Response · Authors · 2024-11-25
>
> Thank you for your constructive feedback. For the mentioned weakness, we want to provide further clarification:
>
> ***Weakness 1:***
> Although RLFH relies on LLM to provide reward signals, it can retrieve external knowledge for factual verification. This task is relatively straightforward for LLM. Additionally, even if a model's response cannot be verified by external knowledge, the LLM can still indicate this by labeling it as "Vague." This ensures that the reward model's signal remains reliable. On the other hand, as an on-policy algorithm, RLFH does not directly inject knowledge into the model but rather corrects the model's use of existing knowledge, reducing the risk of error accumulation. This is one of the advantages of RLFH compared to some other works [1,2].
>
> - [1] Fine-tuning Language Models for Factuality
> - [2] FLAME: Factuality-Aware Alignment for Large Language Models
>
> ***Weakness 2:***
> As shown in Table 1, our method shows improvement over FACT [1], a DPO-based RLHF algorithm which is specifically designed for hallucination, on both ID and OOD evaluation.
>
> - [1] Fine-tuning Language Models for Factuality
>
> ***Weakness 3:***
> Thank you for your reminder. We have made the revisions in the latest version of the paper.

---

> ### Author Response · Authors · 2024-11-28
>
> Dear Reviewer,
>
> We sincerely appreciate your comments, as well as the time you've dedicated to reviewing our work. With the rebuttal deadline approaching, we look forward to any further feedback you might have. Please feel free to reach out with any additional questions or concerns.

---

> > ### Comment · Reviewer_rYQp · 2024-12-03
> > **Thanks for the responses**
> >
> > Actually, my concern is not well addressed by the responses, so I keep my score.

---

### Official Review · Reviewer_xPnr · 2024-11-07

**Soundness:** 2
**Presentation:** 2
**Contribution:** 2
**Rating:** 3
**Confidence:** 3

**Summary:**

The paper, proposes a novel approach called RLFH (Reinforcement Learning for Hallucination) to reduce hallucinations in Large Language Models (LLMs) by fine-grained, token-level  reward. RLFH leverages an online reinforcement learning framework and was tested across multiple datasets and showed improved fact-based accuracy compared to existing hallucination mitigation strategies.

**Strengths:**

1. **Fine-Grained Feedback:** RLFH's use of atomic fact decomposition and token-level rewards enables precise correction of hallucinated responses, surpassing the efficacy of previous, more coarse-grained approaches.

2. **Automated Fact Assessment:** The model employs an LLM-driven feedback mechanism to evaluate truthfulness and informativeness, potentially eliminating human intervention and thus increasing scalability.

**Weaknesses:**

1. **Unsuitable Baselines:** The paper selects "advanced aligned models" such as Zephyr, Orca, Llama2 Chat, and Vicuna 1.5 as baselines (line 311). However, this comparison is unsuitable, as these models differ not only in alignment methods but also in their underlying architectures and training parameters. For a fair comparison, the authors should have evaluated different alignment and hallucination mitigation methods on a consistent model architecture rather than comparing across varied pre-trained models.

2. **Unsignificant effect:** Despite the methodological novelty, the empirical results indicate only marginal improvements in factuality compared to some baseline methods. The reported gains, although statistically significant, may not justify the increased computational complexity and reinforcement learning setup of RLFH, especially given the modest FactScore improvements (e.g., +2.0% on some benchmarks). This raises questions about the practical advantages of RLFH over simpler, less resource-intensive approaches.

3. **Dependence on External Knowledge:** The model’s reliance on external datasets and existing knowledge boundaries raises concerns about its adaptability in real-world scenarios where such data may not be readily available or up-to-date.

**Questions:**

Could the authors replace the current baseline models with a consistent model architecture and compare only different alignment and hallucination mitigation methods?
 This would provide a clearer evaluation of RLFH’s effectiveness by removing confounding factors related to model architecture and pre-training differences.

---

> ### Author Response · Authors · 2024-11-25
>
> Thank you for your helpful feedback. For the mentioned weakness, we would like to provide further clarification:
>
> ***Weakness 1:***
> In selecting baselines, we primarily chose two categories: 1) several advanced aligned models that typically have not undergone specific hallucination mitigation training, making them suitable for demonstrating the effectiveness of hallucination mitigation algorithms; and 2) other hallucination mitigation methods, such as ITI, DOLA, and FACT. We implemented these algorithms using Vicuna-7B and compared them with RLFH, which is also based on Vicuna-7B. Our results demonstrate that, under the premise of a consistent model architecture, our algorithm outperforms other hallucination mitigation algorithms.
>
> ***Weakness 2:***
> To further demonstrate the improvements of our algorithm, we incorporated FAVA [1] as an additional hallucination metric. Unlike FactScore, FAVA detects various types of hallucination errors based on reference documents. We report the average number of errors per sentence as a measure of hallucination:
>
> | Method   | HotpotQA | SquadQA | Biography |
> |----------|----------|---------|-----------------|
> | Origin   | 0.398    | 0.152   | 0.360           |
> | ITI      | 0.218    | 0.310   | 0.239           |
> | Dola     | 0.355    | 0.135   | 0.335           |
> | FACT     | 0.264    | 0.109   | 0.278           |
> | RLFH     | **0.208**    | **0.094**   | **0.232**           |
>
> Compared to FACT and other baselines, RLFH significantly reduces the average error rate, as indicated by errors per sentence. This further validates the effectiveness of our algorithm.
>
> While metrics like FactScore and FAVA focus on hallucination, some hallucination mitigation strategies, such as knowledge editing [2] or training-based methods [3], can inevitably compromise the model's core capabilities. In contrast, RLFH employs an on-policy framework with fine-grained rewards, allowing it to tailor model behavior precisely while minimizing the impact on other capabilities. Below is the evaluation on the Math and Code datasets:
>
> | Method  | GSM8K    | HumanEval |
> |---------|----------|-----------|
> | Origin  | 22.52    | 17.07     |
> | FACT    | 22.37 (-) | 10.98 (-) |
> | RLFH    | 22.90 (+) | 17.68 (+) |
>
> As shown in the table, the RLFH algorithm reduces hallucinations without compromising the performance of the original model. In contrast, FACT reduces performance in areas beyond hallucination, such as Math (evaluated with GSM8K) and Code (evaluated with HumanEval), highlighting an additional advantage of our algorithm.
>
> - [1] Fine-grained Hallucination Detection and Editing for Language Models
> - [2] SelfCheckGPT: Zero-Resource Black-Box Hallucination Detection for Generative Large Language Models
> - [3] Fine-tuning Language Models for Factuality
>
> ***Weakness 3:***
> Although RLFH relies on external knowledge sources for factual verification, we believe this is an important feature to ensure reliable hallucination mitigation. Moreover, as an alignment algorithm, RLFH does not inject new knowledge; instead, it adjusts how the model utilizes its existing internal knowledge. Therefore, in scenarios where external knowledge validation is not possible, we can train the alignment on datasets that allow for such validation and generalize to scenarios without external knowledge bases. The feasibility of this approach is demonstrated in Table 1, where we trained our model on HotpotQA, and its factual accuracy improvements generalized to the out-of-domain test sets SQuADv2 and Biography.
>
> ***Question 1:***
> Thank you for your question. We address this in our response to Weakness 1.

---

> ### Author Response · Authors · 2024-11-28
>
> Dear Reviewer,
>
> We are truly grateful for the feedback you have provided on our work. We have tried our best to address your concerns. With the rebuttal deadline approaching, we are eager to receive any further comments you may have. Please do not hesitate to contact us if you have any additional questions regarding our responses.

---

### Comment · Area_Chair_ivUq · 2024-11-25

Dear Reviewers,

The rebuttal discussion period is coming to a close and the paper currently has a mix of positive and negative reviewers. The authors have spent a lot of time responding to each concern -- can you take a look at the author responses and let them know any remaining concerns you have?

Best,
AC

---

### Meta-Review · Area_Chair_ivUq · 2024-12-22

**Metareview:**

The paper proposes Reinforcement Learning for Hallucination (RLFH), a method to reduce hallucination in large language models (LLMs) by using token-level rewards and on-policy optimization. It breaks down responses into atomic facts, assigns token-level feedback, and employs an automated fact assessment framework for real-time learning. Experiments on benchmarks such as HotpotQA, SQuADv2, and Biography show modest but statistically good improvements in factuality. RLFH's token-level feedback provides more precision than existing methods, and its on-policy optimization aligns rewards with model behavior without requiring human intervention. However, reviewers pointed out issues with inconsistent baselines, small factuality improvements that might not be worth the added computational cost, reliance on external knowledge that affects scalability. While the authors provided a thoughtful rebuttal, the reviewers still have concern about the paper, leading to a rejection decision. Reviewers recommend refining the paper to address these issues for future submission.

**Additional Comments On Reviewer Discussion:**

Reviewers had concerns about the choice of baselines, the lack of RLHF methods in the comparisons, and the limited improvements shown in the results. The authors responded by clarifying some baseline issues and adding new metrics, but these explanations didn’t fully address the reviewers’ concerns. Questions about the reliability of the feedback mechanism and the method’s failure to handle certain types of hallucinations were left unresolved. While minor writing issues were fixed, the paper still need more efforts on presentation.

---

### Decision · Program_Chairs · 2025-01-22

Reject